# Absence of electron-transfer-associated changes in the time-dependent X-ray free-electron laser structures of the photosynthetic reaction center

**Gai Nishikawa[1], Yu Sugo[1], Keisuke Saito[1,2], Hiroshi Ishikita[1,2]***

[1]Department of Applied Chemistry, The University of Tokyo, Tokyo, Japan; [2]Research Center for Advanced Science and Technology, The University of Tokyo, Tokyo, Japan

**\*For correspondence:**
hiro@appchem.t.u-tokyo.ac.jp

**Competing interest:** The authors declare that no competing interests exist.

**Abstract** Using the X-ray free-electron laser (XFEL) structures of the photosynthetic reaction center from *Blastochloris viridis* that show light-induced time-dependent structural changes (Dods et al., (2021) Nature *589*, 310–314), we investigated time-dependent changes in the energetics of the electron-transfer pathway, considering the entire protein environment of the protein structures and titrating the redox-active sites in the presence of all fully equilibrated titratable residues. In the dark and charge separation intermediate structures, the calculated redox potential ($E_m$) values for the accessory bacteriochlorophyll and bacteriopheophytin in the electron-transfer-active branch ($B_L$ and $H_L$) are higher than those in the electron-transfer-inactive branch ($B_M$ and $H_M$). However, the stabilization of the charge-separated $[P_L P_M]^{\bullet+} H_L^{\bullet-}$ state owing to protein reorganization is not clearly observed in the $E_m(H_L)$ values in the charge-separated 5 ps ($[P_L P_M]^{\bullet+} H_L^{\bullet-}$ state) structure. Furthermore, the expected chlorin ring deformation upon formation of $H_L^{\bullet-}$ (saddling mode) is absent in the $H_L$ geometry of the original 5 ps structure. These findings suggest that there is no clear link between the time-dependent structural changes and the electron-transfer events in the XFEL structures.

## eLife assessment

The manuscript describes **valuable** theoretical calculations focusing on the structural changes in the photosynthetic reaction center postulated by others based on time-resolved crystallography using XFEL (Dods *et al.*, Nature, 2021). The authors provide **solid** arguments that calculated changes in redox potential $E_m$ and deformations using the XEFL structures may reflect experimental errors or data processing artifacts rather than real structural changes.

## Introduction

Photosynthetic reaction centers from purple bacteria (PbRC) are heterodimeric reaction centers, which are formed by the protein subunits L and M (*Figure 1*). In PbRC from *Blastochloris viridis*, the electronic excitation of the bacteriochlorophyll *b* (BChl*b*) pair, $[P_L P_M]$, leads to electron transfer to accessory BChl*b*, $B_L$, followed by electron transfer via bacteriopheophytin *b* (BPheo*b*), $H_L$, to menaquinone, $Q_A$, along the electron-transfer active L-branch (A-branch) (*Deisenhofer et al., 1985*). Electron transfer further proceeds from $Q_A$ to ubiquinone, $Q_B$, which is coupled with proton transfer via charged and polar residues in the $Q_B$ binding region (*Rabenstein et al., 1998*). Although the counterpart M-branch (B-branch) is essentially electron-transfer inactive, mutations of the Phe-L181/Tyr-M208 pair to tyrosine/phenylalanine lead to an increase in the yield of $[P_L P_M]^{\bullet+} H_M^{\bullet-}$ formation (~30%), which suggests that these residues are responsible for the energetic asymmetry in the electron-transfer

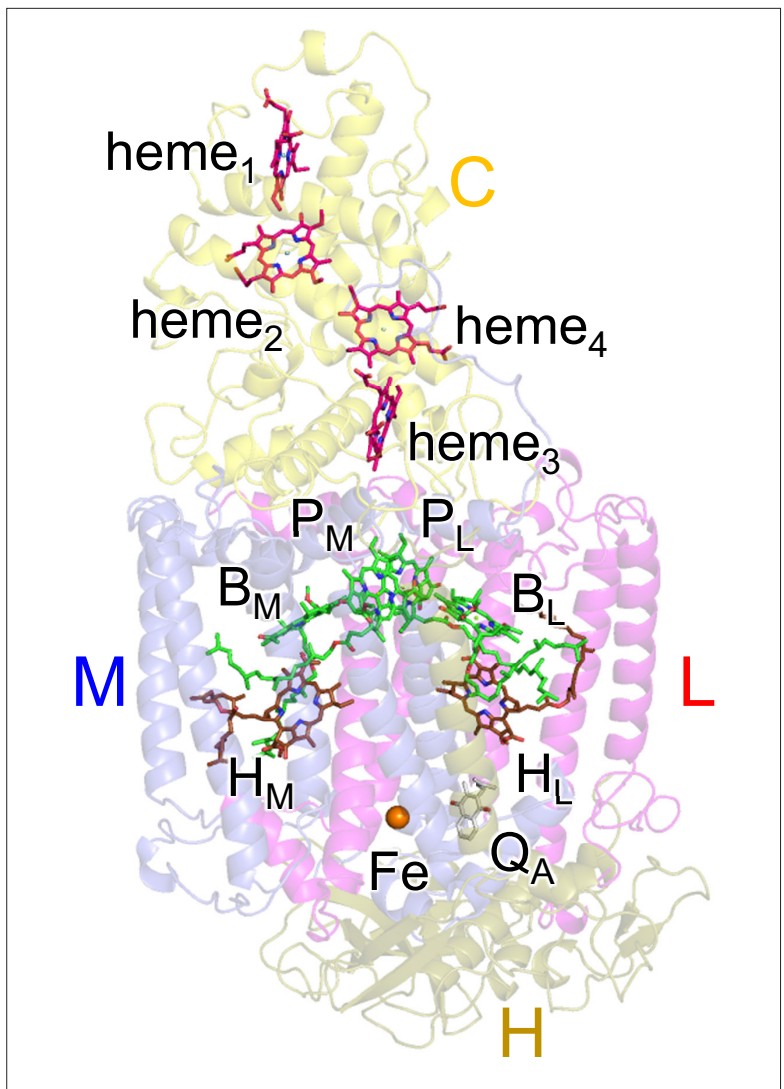

**Figure 1.** Electron-transfer pathways along the L- and M-branches in PbRC from *B. viridis*. The PbRC is composed of the L (red), M (blue), H (gold), and C (yellow) subunits. [$P_L P_M$]: BChl*b* pair; $B_L$ and $B_M$: accessory BChl*b*; $H_L$ and $H_M$: BPheo*b*; $Q_A$: primary quinone (menaquinone); Fe: non-heme Fe complex.

branches (e.g., ***Kirmaier et al., 2003***). The anionic states $B_L^{\bullet-}$, $H_L^{\bullet-}$, and $Q_A^{\bullet-}$ form in ~3.5 ps, ~5 ps, and ~200 ps upon the formation of the electronically excited [$P_L P_M$]* state, respectively (***Holzapfel et al., 1990***). The anionic state formation induces not only reorganization of the protein environment (***Marcus and Sutin, 1985***) but also out-of-plane distortion of the chlorin ring (***Saito et al., 2012***). Two distinct conformations of $H_L^{\bullet-}$ were reported in spectroscopic studies of PbRC from *Rhodobacter sphaeroides* (***Müh et al., 1998***).

Recently, using the X-ray free-electron laser (XFEL), light-induced electron density changes and structural changes of PbRC were analyzed at 1 ps, 5 ps, 20 ps, 300 ps, and 8 µs upon the electronic excitation of [$P_L P_M$] at 960 nm (***Dods et al., 2021***): the 1 ps XFEL structure represents the [$P_L P_M$]* state, the 5 ps and 20 ps XFEL structures represent the charge-separated [$P_L P_M$]$^{\bullet+}H_L^{\bullet-}$ state, and the 300 ps and 8 µs XFEL structures represent the charge-separated [$P_L P_M$]$^{\bullet+}Q_A^{\bullet-}$ state. According to ***Dods et al., 2021***, these XFEL structures revealed how the charge separation process was stabilized by protein conformational dynamics. However, the conclusions drawn from these XFEL structures are based on data with limited resolution. Specifically, eight out of nine XFEL structures have a relatively low resolution of 2.8 Å (atomic coordinates from PDB codes: 5O4C, 6ZI4, and 6ZI5 for dataset a and 6ZHW, 6ZID, 6ZI6, 6ZI9, and 6ZIA for dataset b) (***Dods et al., 2021***). In addition, the data statistics may

indicate that the high-resolution range of some XFEL datasets exhibits high levels of noise (e.g., low $CC_{1/2}$). These observations raise concerns about the reliable comparison of subtle conformational changes among these XFEL structures. Hence, caution must be exercised when interpreting these XFEL structures in terms of their ability to accurately capture relevant conformational changes.

Here, we investigated how the redox potential ($E_m$) values of the BChl$b$ and BPheo$b$ cofactors for one-electron reduction change as electron transfer proceeds using the dark (0 ps), 1 ps, 5 ps, 20 ps, 300 ps, and 8 µs XFEL structures, solving the linear Poisson-Boltzmann equation, and considering the protonation states of all titratable sites in the entire protein. Structural changes (e.g., side-chain reorientation) in the protein environment can be analyzed in the $E_m$ shift, as $E_m$ is predominantly determined by the sum of the electrostatic interactions between the redox-active site and all other groups (i.e., residues and cofactors) in the protein structure. Subtle structural changes of the BChl$b$ and BPheo$b$ chlorin rings, which may not be pronounced even in the $E_m$ shift (*Saito et al., 2012*), can be analyzed in the out-of-plane distortion of the chlorin rings using a normal-coordinate structural decomposition (NSD) analysis (*Jentzen et al., 1997*; *Shelnutt et al., 1998*) with a combination of a quantum mechanical/molecular mechanical (QM/MM) approach in the entire PbRC protein environment.

## Results and discussion
### Energetically asymmetric electron-transfer branches

The XFEL structures show that the $E_m$ values for $B_L$ are ~50 mV higher than those for $B_M$, which facilitates the formation of the charge-separated $[P_L P_M]^{\bullet+} B_L^{\bullet-}$ state and thereby electron transfer along the L-branch (*Figures 2 and 3*). As the $E_m$ profile is substantially consistent with the $E_m$ profile for PbRC from *R. sphaeroides* (*Kawashima and Ishikita, 2018*), it seems plausible that the charge-separated $[P_L P_M]^{\bullet+} B_L^{\bullet-}$ and $[P_L P_M]^{\bullet+} H_L^{\bullet-}$ states in the active L-branch are energetically lower than the $[P_L P_M]^{\bullet+} B_M^{\bullet-}$ and $[P_L P_M]^{\bullet+} H_M^{\bullet-}$ states in the inactive M-branch, respectively, as demonstrated in QM/MM calculations (*Tamura et al., 2020*). Indeed, the calculated $E_m$ values are largely correlated with the lowest unoccupied molecular orbital (LUMO) levels calculated using a QM/MM approach, as suggested previously (coefficient of determination $R^2$=0.98, *Figure 2—figure*

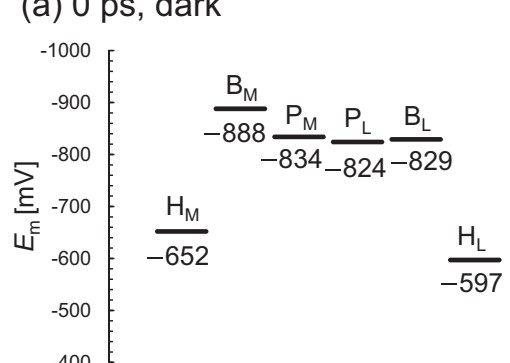

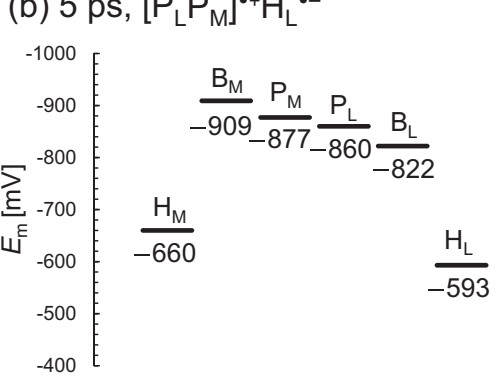

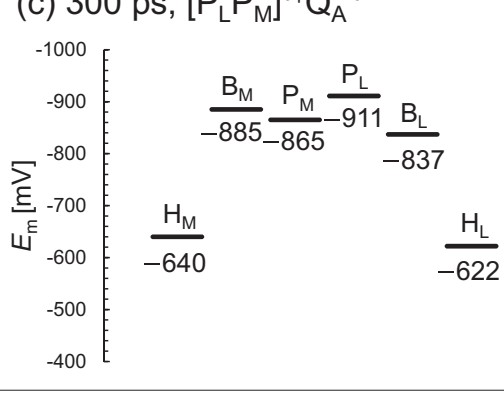

**Figure 2.** $E_m$ profiles in the XFEL structures for dataset a. (**a**) 0 ps. (**b**) 5 ps. (**c**) 300 ps.

The online version of this article includes the following source data and figure supplement(s) for figure 2:

**Figure supplement 1.** $E_m$ values calculated solving the linear Poisson-Boltzmann equation and LUMO energy levels calculated using a QM/MM approach in the dark-state structure.

**Figure supplement 1—source data 1.** Numerical source data for *Figure 2—figure supplement 1*.

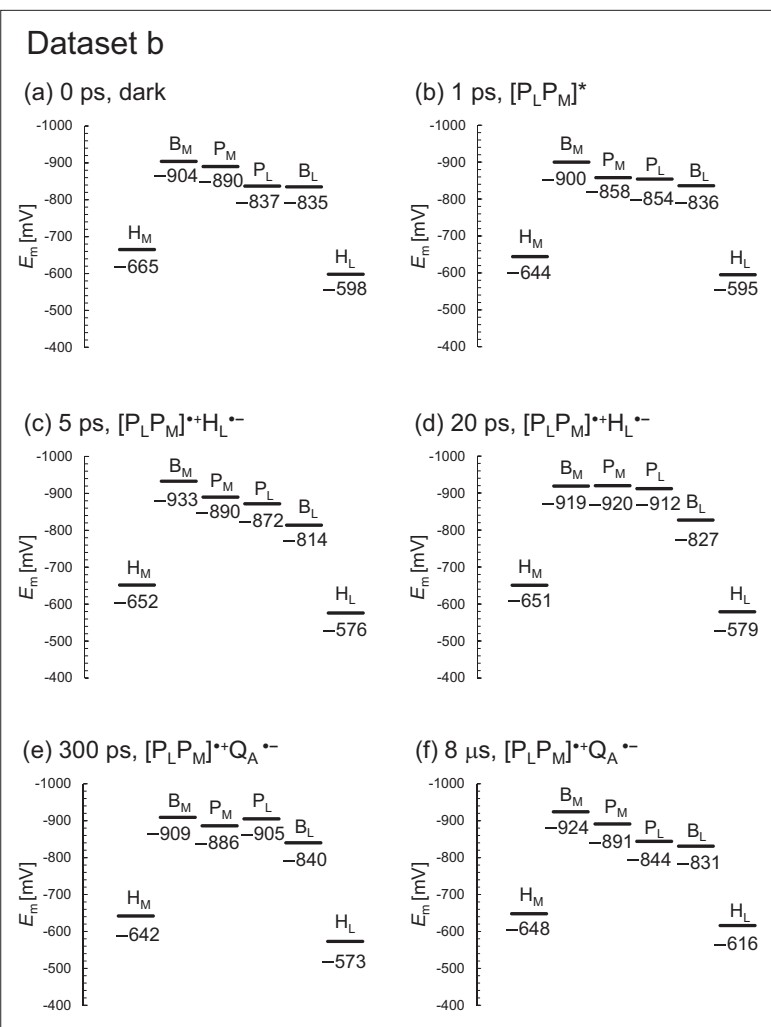

**Figure 3.** $E_m$ profiles in the XFEL structures for dataset b. (**a**) 0 ps. (**b**) 1 ps. (**c**) 5 ps. (**d**) 20 ps. (**e**) 300 ps. (**f**) 8 μs.

*supplement 1*). The $E_m(H_L)$ value of –597 mV (in dataset a; –598 mV in dataset b) is in line with the experimentally estimated value of ca. –600 mV for $H_L$ in PbRC from *B. viridis* (*Rutherford et al., 1979*).

Among the L/M residue pairs, the Phe-L181/Tyr-M208 pair contributes to $E_m(B_L)>E_m(B_M)$ most significantly (25 mV), facilitating L-branch electron transfer, as suggested in theoretical studies (*Gunner et al., 1996*; *Table 1*, *Figure 2*; *Figure 3*; *Figure 4*). This result is also consistent with the contribution of the Phe-L181/Tyr-M210 pair to the difference between $E_m(B_L)$ and $E_m(B_M)$, which was the largest in PbRC from *R. sphaeroides* (*Parson et al., 1990*) (26 mV; *Kawashima and Ishikita, 2018*). The Asn-L158/Thr-M185 pair also contributes to the difference between $E_m(B_L)$ and $E_m(B_M)$ (12 mV, *Table 1*), as does the Val-L157/Thr-M186 pair in PbRC from *R. sphaeroides* (22 mV; *Kawashima and Ishikita, 2018*).

**Table 1.** Contributions of the L/M residue pairs that are responsible for $E_m(B_L)>E_m(B_M)$ (more than 10 mV) in the dark-state structure (mV).
Difference: [contribution of subunit L to $E_m(B_L)$] + [contribution of subunit M to $E_m(B_L)$] – [contribution of subunit L to $E_m(B_M)$] – [contribution of subunit M to $E_m(B_M)$].

| Subunit L | $E_m(B_L)$ | $E_m(B_M)$ | Subunit M | $E_m(B_L)$ | $E_m(B_M)$ | Difference |
|---|---|---|---|---|---|---|
| Phe-L181 | 0 | 17 | Tyr-M208 | 39 | –3 | 25 |
| His-L144 | –8 | –2 | Glu-M171 | –14 | –45 | 25 |
| Asn-L158 | 5 | –6 | Thr-M185 | –3 | –4 | 12 |

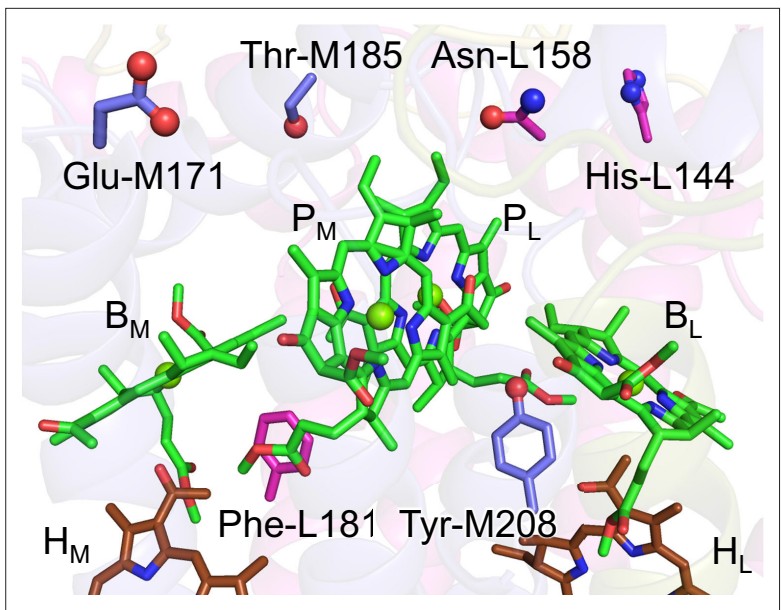

**Figure 4.** Residue pairs that are responsible for $E_m(B_L)>E_m(B_M)$.

For dataset b, the $E_m$ values for $H_L$ are >50 mV higher than those for $H_M$, as observed in $E_m(B_L)$ and $E_m(B_M)$ (*Figure 3*). However, the $E_m$ difference decreases to ~30 mV in the 8 μs XFEL structure (*Figure 3f*), which implies that the 8 μs XFEL structure is distinct from the other XFEL structures (see below). Below, we discuss the dark-state structure if not otherwise specified.

The Ala-L120/Asn-M147 pair contributes to $E_m(H_L)>E_m(H_M)$ most significantly (38 mV) (*Table 2*, *Figure 5*). However, this holds true only for PbRC from *B. viridis*, as Asn-M147 is replaced with alanine (Ala-M149) in PbRC from *R. sphaeroides*. The Asp-L218/Trp-M252 pair decreases $E_m(H_M)$ with respect to $E_m(H_L)$, thereby contributing to $E_m(H_L)>E_m(H_M)$ (20 mV) (*Table 2*; *Figure 5*). Arg-L103 orients toward the protein interior, whereas Arg-M130 orients toward the protein exterior (*Figure 5*), which contributes to $E_m(H_L)>E_m(H_M)$ (17 mV) (*Table 2*). Ser-M271 forms an H-bond with Asn-M147 near $H_M$ (*Figure 5*). Thus, the contribution of Ser-M271 to $E_m(H_L)$ is large, although this residue is replaced with alanine (Ala-M273) in PbRC from *R. sphaeroides*.

## Relevance of structural changes observed in XFEL structures

According to Dods et al., the 5 ps and 20 ps structures correspond to the charge-separated $[P_LP_M]^{\bullet+}H_L^{\bullet-}$ state (*Dods et al., 2021*). If this is the case, $E_m(H_L)$ is expected to be exclusively higher in the 5 ps and 20 ps structures than in the other XFEL structures due to the stabilization of the $[P_LP_M]^{\bullet+}H_L^{\bullet-}$ state

**Table 2.** Contributions of the L/M residue pairs that are responsible for $E_m(H_L)>E_m(H_M)$ (more than 10 mV) in the dark-state structure (mV).
Difference: [contribution of subunit L to $E_m(H_L)$] + [contribution of subunit M to $E_m(H_L)$] – [contribution of subunit L to $E_m(H_M)$] – [contribution of subunit M to $E_m(H_M)$].

| Subunit L | $E_m(H_L)$ | $E_m(H_M)$ | Subunit M | $E_m(H_L)$ | $E_m(H_M)$ | Difference |
|-----------|-----------|-----------|-----------|-----------|-----------|------------|
| Ala-L120 | –4 | 0 | Asn-M147 | 0 | –42 | 38 |
| Asp-L218 | –2 | –22 | Trp-M252 | 1 | 0 | 20 |
| Arg-L103 | 77 | 3 | Arg-M130 | 3 | 59 | 17 |
| Ala-L237 | –2 | 0 | Ser-M271 | 3 | –16 | 16 |
| Lys-L110 | 17 | 2 | Ala-M137 | 0 | 3 | 14 |
| Val-L219 | 1 | 5 | Thr-M253 | 17 | 1 | 11 |
| His-L211 | 1 | 0 | Arg-M245 | 14 | 4 | 11 |

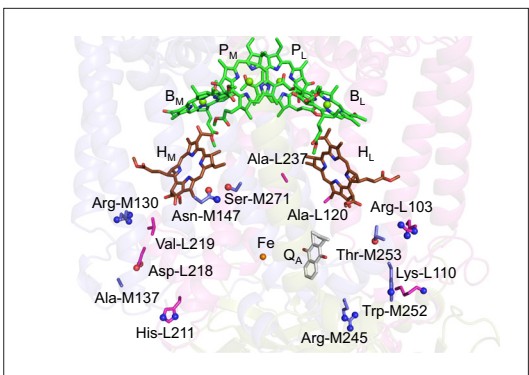

**Figure 5.** Residue pairs that are responsible for $E_m(H_L) > E_m(H_M)$.

by protein reorganization. In dataset a, the $E_m(H_L)$ value is only 4 mV higher in the 5 ps structure than in the dark structure (*Figure 6a*). In dataset b, the $E_m(H_L)$ value is ~20 mV higher in the 5 ps and 20 ps structures than in the dark structure (*Figure 6b*). However, the $E_m(H_L)$ value is 25 mV higher in the 300 ps structure than in the dark structure. *Tables 3 and 4* show the residues that contribute to the slight increase in $E_m(H_L)$ most significantly in the 5 ps and 20 ps structures. Most of these residues were in the region where Dods et al. specifically performed multiple rounds of partial occupancy refinement (e.g., 153–178, 190, 230, and 236–248 of subunit L and 193–221, 232,

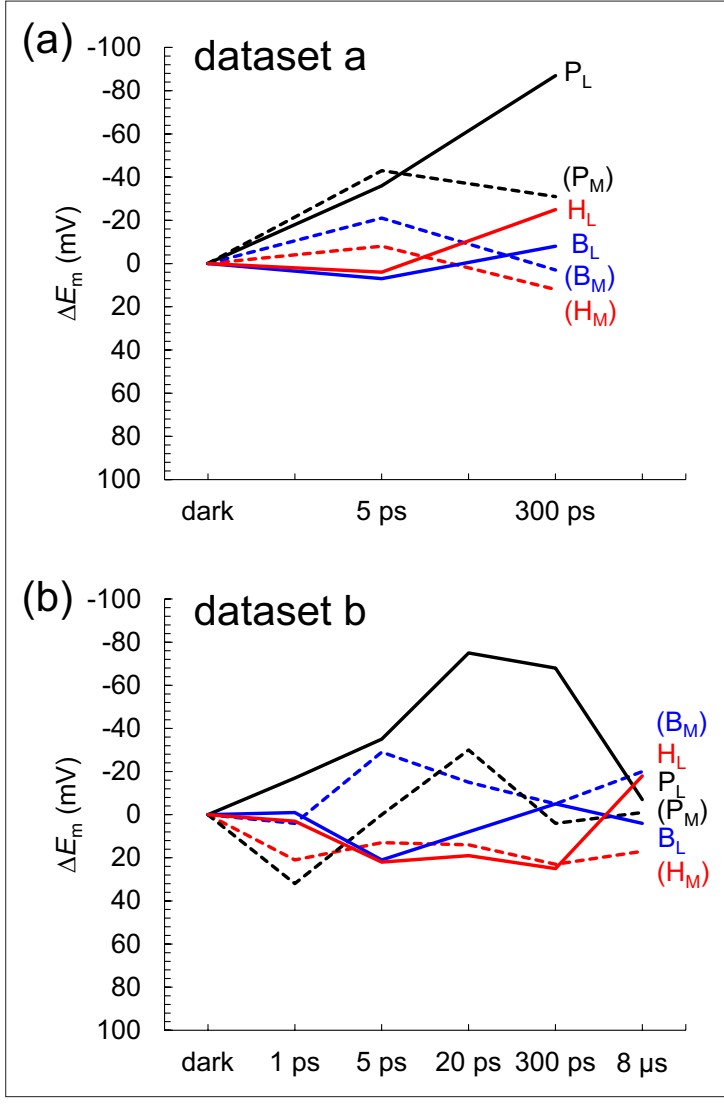

**Figure 6.** Time-dependent $E_m$ changes for BChl*b* and BPheo*b* in the XFEL structures. (**a**) Dataset a. (**b**) Dataset b. $\Delta E_m$ denotes the $E_m$ shift with respect to the dark-state structure. Black solid lines: $P_L$; black dotted lines: $P_M$; blue solid lines $B_L$; blue dotted lines: $B_M$; red solid lines: $H_L$; red dotted lines: $H_M$.

The online version of this article includes the following source data for figure 6:

**Source data 1.** Numerical source data for *Figure 6*.

**Table 3.** Residues that shift $E_m(H_L)$ most significantly during putative electron transfer in the XFEL structures (dataset a) (mV).

The same residues are highlighted in the same colors for clarity.

| Dataset a | | Shift | | Shift |
|---|---|---|---|---|
| 0–5 ps | Ser-L176 | 5 | Cys-M210 | 4 |
| | Thr-M220 | −7 | $B_L$ | −5 |
| 5–300 ps | $B_L$ | 7 | Gly-M209 | 3 |
| | Gly-M211 | −11 | Leu-M212 | −8 |

243–253, 257–266 of subunit M) (**Dods et al., 2021**). In dataset b (**Table 4**), which has more data points than dataset a (**Table 3**), the contributions of these residues to $E_m(H_L)$ often fluctuate (e.g., upshift/downshift followed by downshift/ upshift) at different time intervals (e.g., 1–5 ps, 5–20 ps, and 20–300 ps). This result suggests that the structural differences among the XFEL structures are not related to the actual time course of charge separation. Furthermore, the $E_m(H_M)$ value in the inactive M-branch is also ~15 mV higher in the 5 ps and 20 ps structures than in the dark structure (**Figure 6b**). These results suggest that the ~20 mV higher $E_m(H_L)$ value in the 5 ps and 20 ps structures is not specifically due to the formation of the $[P_L P_M]^{•+} H_L^{•−}$ state. Thus, the stabilization of the $[P_L P_M]^{•+} H_L^{•−}$ state owing to protein reorganization is not clearly observed in the $E_m(H_L)$ values.

An NSD analysis (**Jentzen et al., 1997**; **Shelnutt et al., 1998**) of the out-of-plane distortion of the chlorin ring is sensitive to subtle structural changes in the chlorin ring, which are not distinct in the $E_m$ changes (**Saito et al., 2012**). QM/MM calculations indicate that $H_L^{•−}$ formation induces the saddling mode in the chlorin ring, which describes the movement of rings I and III being in the opposite direction to the movement of rings II and IV along the normal axis of the chlorin ring (**Tables 5 and 6**). However, (i) in the XFEL structures, the saddling mode of $H_L$ remains practically unchanged in dataset a during electron transfer (**Figure 7** and **Supplementary files 1 and 2**). In dataset b, the saddling mode of $H_L$ is induced most significantly at 1 ps, which does not correspond to the charge-separated $[P_L P_M]^{•+} H_L^{•−}$ state (**Figure 8**). (ii) In addition, the ruffling mode is more pronounced than the saddling mode in $H_L$ (**Figure 8**), which suggests that the observed deformation of $H_L$ is not directly associated with the reduction of $H_L$.

**Table 4.** Residues that shift $E_m(H_L)$ most significantly during putative electron transfer in the XFEL structures (dataset b) (mV).

The same residues are highlighted in the same colors for clarity.

| Dataset b | | Shift | | Shift |
|---|---|---|---|---|
| 0–1 ps | Ser-L238 | 8 | Ser-L176 | 7 |
| | $B_L$ | −7 | Leu-M213 | −3 |
| 1–5 ps | Gly-M211 | 6 | Leu-M213 | 5 |
| | Ser-L238 | −6 | Thr-M253 | −5 |
| 5–20 ps | $B_L$ | 12 | Thr-M253 | 7 |
| | Leu-M213 | −4 | $P_M$ | −3 |
| 20–300 ps | Ser-L238 | 3 | Gly-M211 | 2 |
| | $B_L$ | −10 | Glu-L212 | −4 |
| 300 ps to 8 µs | Glu-L212 | 4 | Leu-M213 | 4 |
| | $B_L$ | −6 | Gly-M211 | −5 |

One might argue that the loss of the link between the formation of the charge-separated state and the $E_m(H_L)$ change (**Figure 6**) is not due to experimental errors, including data processing artifacts, but rather represents the actual ps timescale phenomena during the primary charge-separation reactions (e.g., Dods et al. noted that 'the primary electron-transfer step to $H_L$ is more rapid than conventional Marcus theory'; **Dods et al., 2021**). Even if this were the case, this hypothesis regarding the relevance of the XFEL structures to the electron-transfer events could be further explored by examining the changes in $E_m(Q_A)$ among the XFEL structures, considering the relatively slow electron-transfer step to $Q_A$ that allows sufficient protein relaxation to occur (e.g., Dods et al. stated that 'the electron-transfer step to $Q_A$ has a single exponential decay time of 230±30 ps, consistent with conventional Marcus theory'; **Dods et al., 2021**). That is, if the $E_m(Q_A)$ values are not higher in the 300 ps and 8 µs structures than in the other structures, it suggests that significant experimental errors exist, rendering the XFEL structures irrelevant to the electron-transfer events. Consistent with this perspective, the present results demonstrate that the $E_m(Q_A)$ values in the 300 ps and 8 µs structures are not

**Table 5.** Induced out-of-plane distortion of $H_L$ and $H_M$ in the PbRC protein environment of the dark structure for dataset a in response to the reduction (Å).

| | Saddling | Ruffling | Doming | Waving | | Propellering |
|---|---|---|---|---|---|---|
| | $B_{2u}$ | $B_{1u}$ | $A_{2u}$ | $E_{g(x)}$ | $E_{g(y)}$ | $A_{1u}$ |
| $H_L$ | 0.18 | 0.35 | –0.10 | 0.13 | –0.11 | 0.13 |
| $H_L{}^{\bullet-}$ | 0.24 | 0.35 | –0.09 | 0.12 | –0.12 | 0.13 |
| $(P_L{}^{\bullet+}H_L{}^{\bullet-})$ | (0.22) | (0.36) | (–0.07) | (0.13) | (–0.13) | (0.13) |
| $H_L/H_L{}^{\bullet-}$ difference | 0.06 | 0.00 | 0.01 | –0.01 | –0.01 | 0.00 |
| | | | | | | |
| $H_M$ | 0.06 | 0.40 | –0.20 | 0.37 | 0.12 | 0.19 |
| $H_M{}^{\bullet-}$ | 0.12 | 0.38 | –0.22 | 0.33 | 0.09 | 0.22 |
| $(P_L{}^{\bullet+}H_M{}^{\bullet-})$ | (0.14) | (0.38) | (–0.22) | (0.33) | (0.10) | (0.22) |
| $H_M/H_M{}^{\bullet-}$ difference | 0.06 | –0.02 | –0.02 | –0.04 | –0.03 | 0.03 |

The online version of this article includes the following source data for table 5:

**Source data 1.** Numerical source data for *Table 5*.

significantly higher than those in the other structures, including the dark-state structure (*Figure 9*). Consequently, the lack of a clear relationship between the charge-separated state and the changes in $E_m(Q_A)$ at 300 ps and 8 µs further strengthens the argument that the XFEL structures are irrelevant to the electron-transfer events.

In summary, the $E_m$ values in the active L-branch are higher than those in the inactive M-branch in the XFEL structures, which suggests that electron transfer via $B_L{}^{\bullet-}$ and $H_L{}^{\bullet-}$ is energetically more favored than that via $B_M{}^{\bullet-}$ and $H_M{}^{\bullet-}$ (*Figure 2*). The Phe-L181/Tyr-M208 pair contributes to the difference between $E_m(B_L)$ and $E_m(B_M)$ the most significantly, as observed in the Phe-L181/Tyr-M210 pair in PbRC from *R. sphaeroides* (*Kawashima and Ishikita, 2018*; *Parson et al., 1990*). The stabilization of the $[P_L P_M]{}^{\bullet+}H_L{}^{\bullet-}$ state owing to protein reorganization is not clearly observed in the $E_m(H_L)$ values (*Figure 6*). The absence of the induced saddling mode in the $H_L$ chlorin ring in the 5 ps and 20 ps structures suggests that $H_L{}^{\bullet-}$ does not specifically exist in these XFEL structures (*Figures 7 and 8*). The cyclic fluctuations in the contributions of the residues to $E_m(H_L)$ at different time intervals suggest that the structural differences among the XFEL structures are not related to the actual time course of charge separation (*Table 4*). The major limitation of the structural studies conducted by *Dods et al., 2021*, is the relatively low resolution of their XFEL structures, primarily at 2.8 Å. Consequently, the

**Table 6.** Induced out-of-plane distortion of $H_L$ and $H_M$ in the PbRC protein environment of the dark structure for dataset b in response to the reduction (Å).

| | Saddling | Ruffling | Doming | Waving | | Propellering |
|---|---|---|---|---|---|---|
| | $B_{2u}$ | $B_{1u}$ | $A_{2u}$ | $E_{g(x)}$ | $E_{g(y)}$ | $A_{1u}$ |
| $H_L$ | 0.13 | 0.35 | –0.13 | 0.07 | –0.09 | 0.20 |
| $H_L{}^{\bullet-}$ | 0.25 | 0.34 | –0.02 | 0.12 | –0.16 | 0.13 |
| $(P_L{}^{\bullet+}H_L{}^{\bullet-})$ | (0.23) | (0.34) | (–0.03) | (0.12) | (–0.16) | (0.12) |
| $H_L/H_L{}^{\bullet-}$ difference | 0.12 | –0.01 | 0.11 | 0.05 | –0.07 | –0.07 |
| $H_M$ | 0.08 | 0.57 | –0.11 | 0.16 | 0.20 | 0.32 |
| $H_M{}^{\bullet-}$ | 0.16 | 0.36 | –0.19 | 0.36 | 0.18 | 0.21 |
| $(P_L{}^{\bullet+}H_M{}^{\bullet-})$ | (0.16) | (0.36) | (–0.20) | (0.36) | (0.18) | (0.21) |
| $H_M/H_M{}^{\bullet-}$ difference | 0.08 | –0.21 | –0.08 | 0.20 | –0.02 | –0.11 |

The online version of this article includes the following source data for table 6:

**Source data 1.** Numerical source data for *Table 6*.

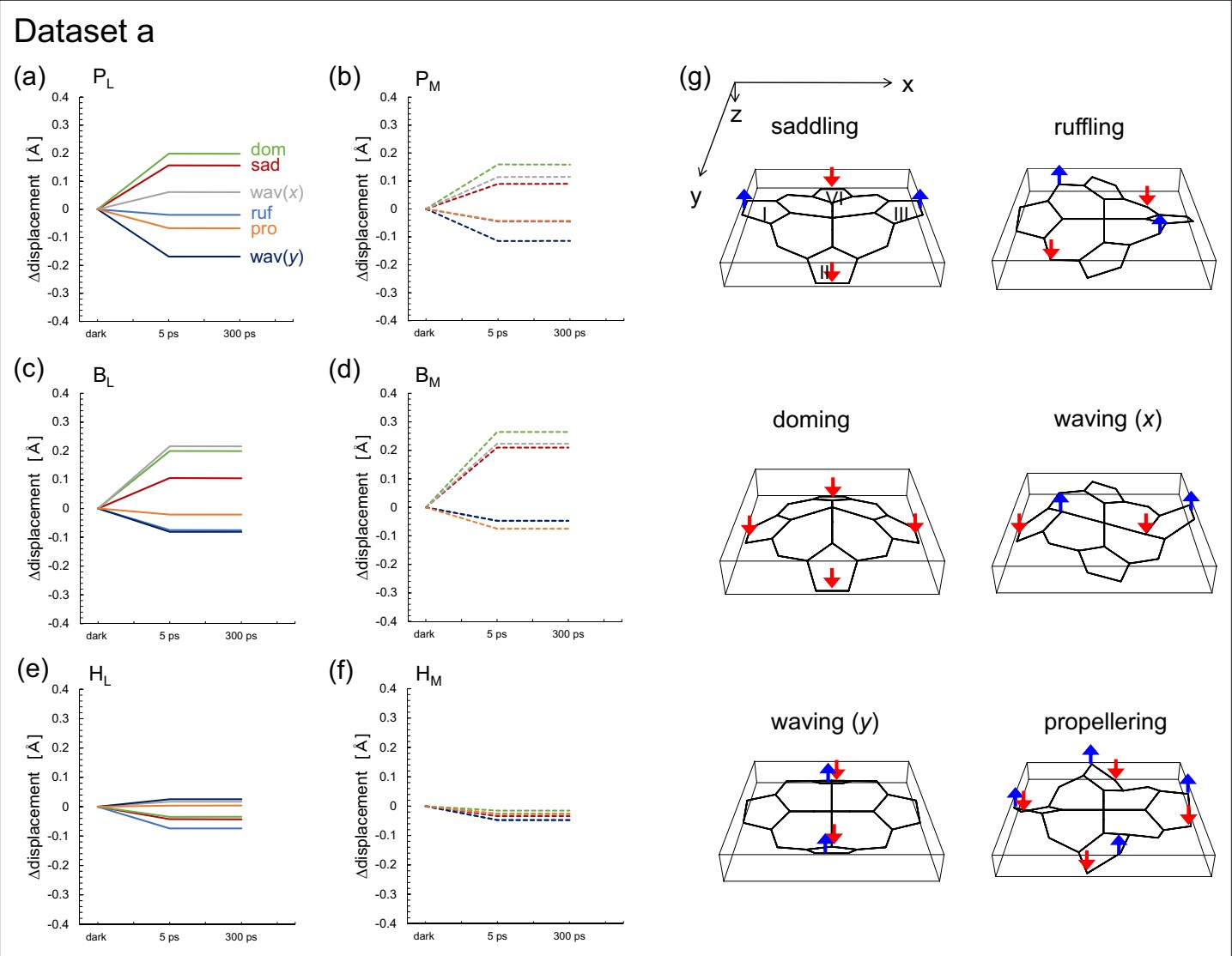

**Figure 7.** Time-dependent changes in the lowest frequency out-of-plane modes of the chlorin rings in the XFEL structures (dataset a). (a) $P_L$. (b) $P_M$. (c) $B_L$. (d) $B_M$. (e) $H_L$. (f) $H_M$. (g) Typical lowest frequency out-of-plane modes of the chlorin rings. Sad: saddling (red); ruf: ruffling (blue); dom: doming (green); wav($x$, $y$): waving ($x$, $y$) (gray, dark blue); pro: propellering (orange). Solid and dotted lines indicate L- and M-branches, respectively. See **Supplementary file 1** for the absolute values in the dark state for dataset a.

The online version of this article includes the following source data for figure 7:

**Source data 1.** Numerical source data for **Figure 7**.

observed changes in $E_m$ values and chlorin ring deformations are more likely to reflect experimental errors or data processing artifacts rather than actual structural changes induced by electron-transfer events. This concern is reinforced by the lack of a clear relationship between the actual $Q_A^{\bullet-}$ formation and the $E_m(Q_A)$ values in the 300 ps and 8 µs structures (**Figure 9**). Consequently, the time-dependent structural changes proposed by **Dods et al., 2021**, are highly likely irrelevant to the electron-transfer events.

Hence, it is crucial to exercise caution when interpreting time-dependent XFEL structures, especially in the absence of comprehensive evaluations of the energetics for accompanying structural changes. This cautionary note should serve as a counterargument in the future, highlighting the potential pitfalls associated with presenting time-dependent XFEL structures of insufficient quality and drawing conclusive interpretations of protein structural changes that may not be distinguishable from significant experimental errors or data processing artifacts. Future high-resolution structures

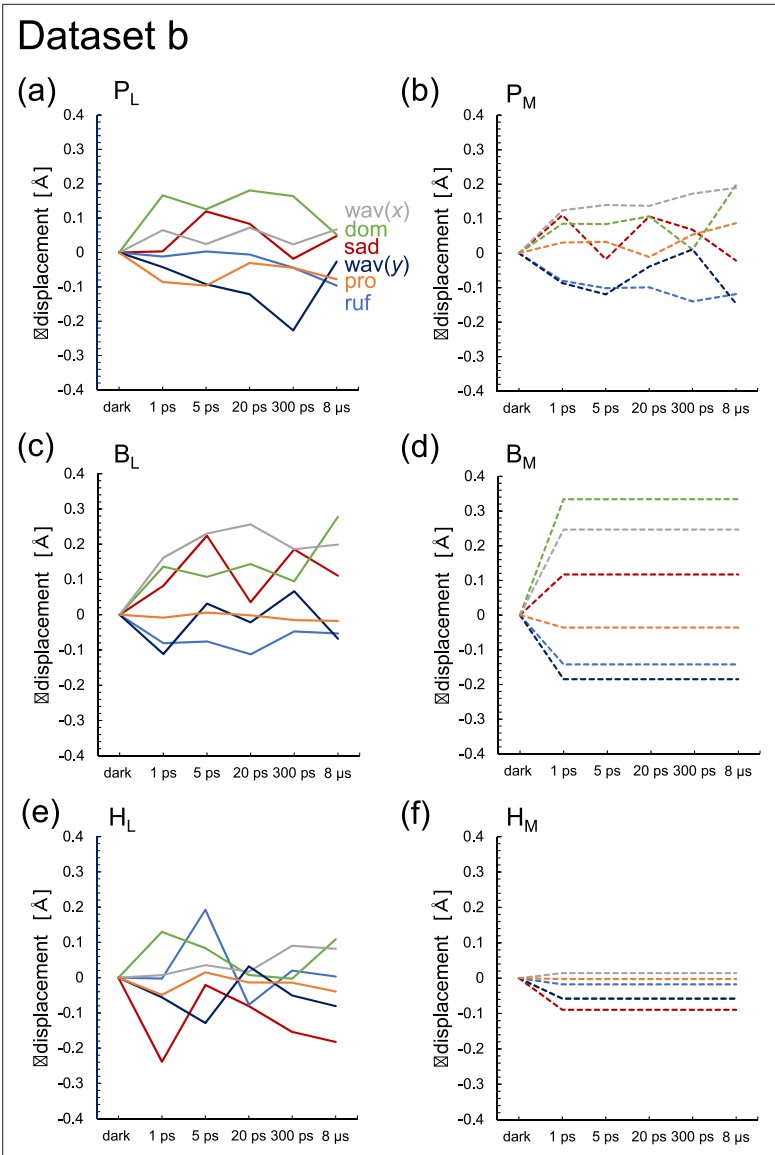

**Figure 8.** Time-dependent changes in the lowest frequency out-of-plane modes of the chlorin rings in the XFEL structures (dataset b). (a) $P_L$. (b) $P_M$. (c) $B_L$. (d) $B_M$. (e) $H_L$. (f) $H_M$. Sad: saddling (red); ruf: ruffling (blue); dom: doming (green); wav(x, y): waving (x, y) (gray, dark blue); pro: propellering (orange). Solid and dotted lines indicate L- and M-branches, respectively. See ***Supplementary file 2*** for the absolute values in the dark state for dataset b ***Figure 8—source data 1***.

The online version of this article includes the following source data for figure 8:

**Source data 1.** Numerical source data for ***Figure 8***.

may provide further insights into the actual structural changes relevant to electron-transfer events. By combining both high-resolution structures and rigorous energetic evaluations, a more comprehensive understanding of the protein structure-function relationship can be achieved.

## Methods
### Coordinates and atomic partial charges

The atomic coordinates of PbRC from *B. viridis* were taken from the XFEL structures determined at 0 ps (dark state; PDB code 5O4C for dataset a and 5NJ4 for dataset b), 1 ps ([$P_LP_M$]* state; PDB code, 6ZHW for dataset b), 5 ps ([$P_LP_M$] •+$H_L$•− state; PDB code, 6ZI4 for dataset a and 6ZID for dataset b),

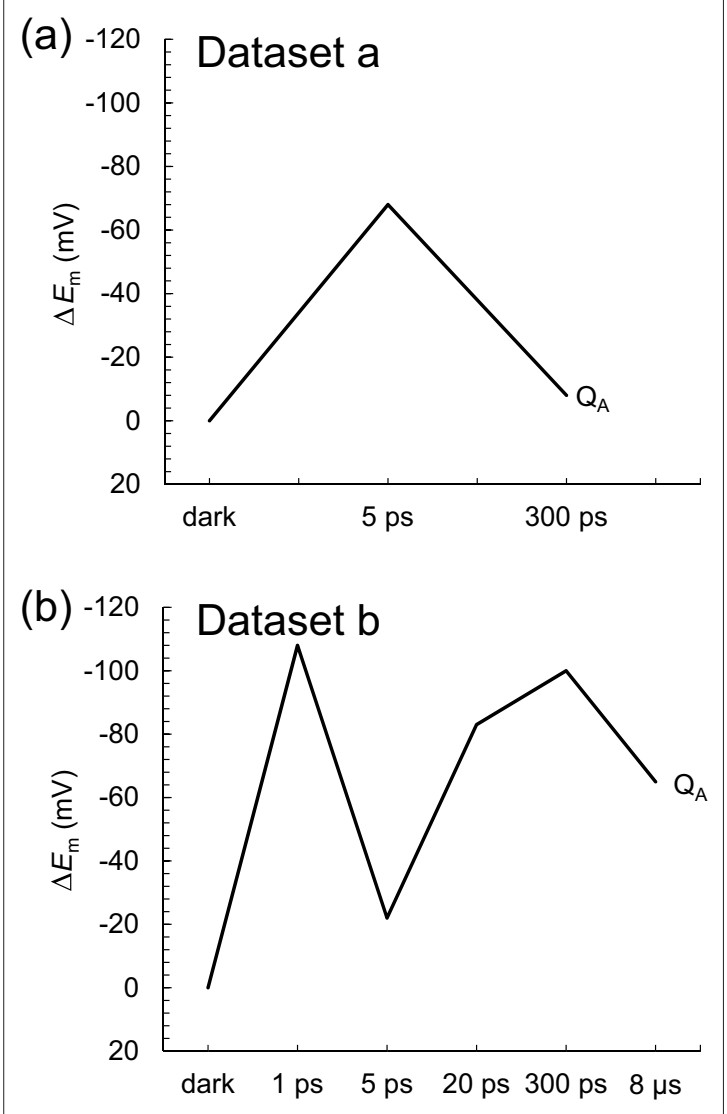

**Figure 9.** Time-dependent $E_m$ changes for $Q_A$ in the XFEL structures. (**a**) Dataset a. (**b**) Dataset b. $\Delta E_m$ denotes the $E_m$ shift with respect to the dark-state structure. Note that the calculated $E_m(Q_A)$ values for dataset a and dataset b in the dark structure are –223 mV and –209 mV, respectively, which are comparable to experimentally measured values of –150 mV for PbRC from *B. viridis* (menaquinone) (**Prince et al., 1976**) and –180 mV for PbRC from *R. sphaeroides* (ubiquinone) (**Prince and Dutton, 1976**).

The online version of this article includes the following source data for figure 9:

**Source data 1.** Numerical source data for **Figure 9**.

20 ps ([$P_L P_M$] $^{\bullet+}H_L^{\bullet-}$ state; PDB code, 6ZI6 for dataset b), 300 ps ([$P_L P_M$] $^{\bullet+}Q_A^{\bullet-}$ state; PDB code, 6ZI5 for dataset a and 6ZI9 for dataset b), and 8 μs ([$P_L P_M$] $^{\bullet+}Q_A^{\bullet-}$ state; PDB code, 6ZIA for dataset b). Atoms with 30% occupancy for the photoactivated state (**Dods et al., 2021**) were used wherever present. Hydrogen atoms were generated and energetically optimized with CHARMM (**Brooks et al., 1983**). The atomic partial charges of the amino acids were obtained from the all-atom CHARMM22 (**MacKerell et al., 1998**) parameter set. For diacylglycerol, the Fe complex (**Kawashima and Ishikita, 2018**), and menaquinone (**Kawashima and Ishikita, 2017**), the atomic charges were adopted from previous studies. The atomic charges of BChl*b* and BPheo*b* (BChl*b*, BChl*b*$^{\bullet+}$, BChl*b*$^{\bullet-}$, BPheo*b*, and BPheo*b*$^{\bullet-}$) were determined by fitting the electrostatic potential in the neighborhood of these molecules using the RESP procedure (**Bayly et al., 1993**; **Supplementary file 3**). The electronic densities were calculated after geometry optimization using the density functional theory (DFT) method with the B3LYP

functional and 6-31G** basis sets in the JAGUAR program (*Jaguar, 2012*). For the atomic charges of the nonpolar CH$_n$ groups in the cofactors (e.g., the phytol chains of BChl*b* and BPheo*b* and the isoprene side chains of quinone), a value of +0.09 was assigned to nonpolar H atoms.

## Calculation of $E_m$: solving the linear Poisson-Boltzmann equation

The $E_m$ values in the protein were determined by calculating the electrostatic energy difference between the two redox states in a reference model system. This was achieved by solving the linear Poisson-Boltzmann equation with the MEAD program (*Bashford and Karplus, 1990*) and using $E_m$(B-Chl*b*) = –665 mV and $E_m$(BPheo*b*) = –429 mV (based on $E_m$(BChl*b*) = –700 mV and $E_m$(BPheo*b*) = –500 mV for one-electron reduction measured in dimethylformamide; *Fajer et al., 1976*; *Watanabe and Kobayashi, 1991*), considering the solvation energy difference. The $E_m$(Q$_A$) value was calculated, using the reference $E_m$ value of –256 mV versus NHE for menaquinone-2 in water (*Kishi et al., 2017*). The difference in the $E_m$ value of the protein relative to the reference system was added to the known $E_m$ value. To account for the ensemble of protonation patterns, a Monte Carlo method with Karlsberg was used for sampling (*Rabenstein and Knapp, 2001*). The linear Poisson-Boltzmann equation was solved using a three-step grid-focusing procedure with resolutions of 2.5 Å, 1.0 Å, and 0.3 Å. Monte Carlo sampling provided the probabilities [A$_{ox}$] and [A$_{red}$] of the two redox states of molecule A, and $E_m$ was evaluated using the Nernst equation. A bias potential was applied to ensure an equal amount of both redox states ([A$_{ox}$] = [A$_{red}$]), thus determining the redox midpoint potential as the resulting bias potential. To ensure consistency with previous computational results, we used identical computational conditions and parameters as previous studies (e.g., *Kawashima and Ishikita, 2018*), performing all computations at 300 K, pH 7.0, and an ionic strength of 100 mM. The dielectric constants were set to 4 for the protein interior and 80 for water.

## QM/MM calculations

We employed the restricted DFT method for describing the closed-shell electronic structure and the unrestricted DFT method for the open-shell electronic structure with the B3LYP functional and LACVP* basis sets using the QSite (*QSite, 2012*) program. To neutralize the entire system, counter ions were added randomly around the protein using the Autoionize plugin in VMD (*Humphrey et al., 1996*). In the QM region, all atom positions were relaxed in the QM region, while the H-atom positions were relaxed in the MM region. The QM regions were defined as follows: for the BChl*b* pair [P$_L$P$_M$]: the side chains of the ligand residues (His-L173 and His-M200) and H-bond partners (His-L168, Tyr-M195, and Thr-L248); for accessory BChl*b*: B$_L$/B$_M$ and the side chain of the ligand residue (His-L153 for B$_L$/His-M180 for B$_M$); for BPheo*b*: H$_L$/H$_M$.

## NSD analysis

To analyze the out-of-plane distortions of chlorin rings, we employed an NSD procedure with the minimal basis approximation, where the deformation profile can be represented by the six lowest-frequency normal modes, that is, ruffling (B$_{1u}$), saddling (B$_{2u}$), doming (A$_{2u}$), waving (E$_{g(x)}$ and E$_{g(y)}$), and propellering (A$_{1u}$) modes (*Jentzen et al., 1997*; *Shelnutt et al., 1998*). The NSD analysis was performed in the following three steps, as performed previously (*Saito et al., 2012*). First, the atomic coordinates of the Mg-substituted macrocycle were extracted from the crystal or QM/MM optimized structures (*Table 5—source data 1*, *Table 6—source data 1*). Second, the extracted coordinates were superimposed on the reference coordinates of the macrocycle. The superimposition is based on a least-square method, and the mathematical procedure is described in *Zucchelli et al., 2007*. Finally, the out-of-plane distortion in the superimposed coordinates was decomposed into the six lowest-frequency normal modes by the projection to the reference normal mode coordinates as

$$d^\Gamma = \sum_{i=1}^{N} \Delta z_i \left( n_z^\Gamma \right)_i, \tag{1}$$

where $d^\Gamma$ represents the distortion component of the mode $\Gamma$ (i.e., $\Gamma$ = B$_{1u}$, B$_{2u}$, A$_{2u}$, E$_{g(x)}$, E$_{g(y)}$, or A$_{1u}$), $\Delta z_i$ is the $z$-component of the superimposed coordinates in the $i$th heavy atom, and $\left( n_z^\Gamma \right)_i$ is the $z$-component of the normalized eigenvector of the reference normal mode $\Gamma$ in the $i$th heavy atom. $N$ represents the number of heavy atoms. See *Saito et al., 2012*, for further details.

## Acknowledgements

This research was supported by JSPS KAKENHI (JP23H04963 to KS; JP20H03217 and JP23H02444 to HI) and Interdisciplinary Computational Science Program in CCS, University of Tsukuba.

## Additional information

### Funding

| Funder | Grant reference number | Author |
| --- | --- | --- |
| Japan Society for the Promotion of Science | JP23H04963 | Keisuke Saito |
| Japan Society for the Promotion of Science | JP20H03217 | Hiroshi Ishikita |
| Japan Society for the Promotion of Science | JP23H02444 | Hiroshi Ishikita |
| University of Tsukuba | Interdisciplinary Computational Science Program in CCS | Keisuke Saito |

The funders had no role in study design, data collection and interpretation, or the decision to submit the work for publication.

### Author contributions

Gai Nishikawa, Validation, Investigation; Yu Sugo, Methodology; Keisuke Saito, Investigation, Methodology; Hiroshi Ishikita, Conceptualization, Supervision, Investigation, Writing - original draft, Writing - review and editing

### Author ORCIDs

Keisuke Saito http://orcid.org/0000-0002-2293-9743
Hiroshi Ishikita http://orcid.org/0000-0002-5849-8150

Reviewer #1 (Public Review): https://doi.org/10.7554/eLife.88955.4.sa1
Reviewer #2 (Public Review): https://doi.org/10.7554/eLife.88955.4.sa2
Author Response https://doi.org/10.7554/eLife.88955.4.sa3

## Additional files

### Supplementary files

• Supplementary file 1. Out-of-plane distortions in the PbRC protein environment of the dark structure for dataset a (Å).

• Supplementary file 2. Out-of-plane distortions in the PbRC protein environment of the dark structure for dataset b (Å).

• Supplementary file 3. Atomic charges of BChl*b* and BPheo*b*.

• MDAR checklist

### Data availability

Figure 2 - figure supplement 1 - source data 1, Figure 6 - source data 1, Figure 7 - source data 1, Figure 8 - source data 1, Figure 9 - source data 1, Table 5 - source data 1, and Table 6 - source data 1 contain the numerical data used to generate the figures and tables.

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
