## [Editor Report · eLife assessment]

The manuscript describes **valuable** theoretical calculations focusing on the structural changes in the photosynthetic reaction center postulated by others based on time-resolved crystallography using XFEL (Dods *et al*., Nature, 2021). The authors provide **solid** arguments that calculated changes in redox potential *E*_m_ and deformations using the XEFL structures may reflect experimental errors or data processing artifacts rather than real structural changes.

---

## [Referee Report · Reviewer #1 (Public Review)]

First, I agree with the authors of this manuscript that conformational changes in the XFEL structures with 2.8 A resolution are not reliable enough for demonstrating the subtle changes in the electron transfer events in this bacterial photosynthesis system. Actually, the data statistics in the paper by Dods et al. showed that the high-resolution range of some of the XFEL datasets may include pretty high noise (low CC1/2 and high Rsplit) so the comparison of the subtle conformational changes of the structures is problematic.

The manuscript by Gai Nishikawa investigated time-dependent changes in the energetics of the electron transfer pathway based on the structures by Dods et al. by calculating redox potential of the active and inactive branches in the structures and found no clear link between the time-dependent structural changes and the electron transfer events in the XFEL structures published by Dods, R.et al. (2021). This study provided validation for the interpretation of the structures of those electron-transferring proteins.

The paper was well prepared.

Comments on latest version:

The revisions the authors have made have improved the manuscript.

---

## [Referee Report · Reviewer #2 (Public Review)]

The manuscript by Nishikawa et al. addresses time-dependent changes in the electron transfer energetics in the photosynthetic reaction center from Blastochloris viridis, whose time-dependent structural changes upon light illumination were recently demonstrated by time-resolved serial femtosecond crystallography (SFX) using X-ray free-electron laser (XFEL) (Dods et al., Nature, 2021). Based on the redox potential Em values of bacteriopheophytin in the electron transfer active branch (BL) by solving the linear Poisson-Boltzmann equation, the authors found that Em(HL) values in the charge-separated 5-ps structure obtained by XFEL are not clearly changed, suggesting that the P+HL- state is not stabilized owing to protein reorganization. Furthermore, chlorin ring deformation upon HL- formation, which was expected from their QM/MM calculation, is not recognized in the 5-ps XFEL structure. Then the authors concluded that the structural changes in the XFEL structures are not related to the actual time course of charge separation. They argued that their calculated changes in Em and chlorin ring deformations using the XEFL structures may reflect the experimental errors rather than the real structural changes; they mentioned this problem is due to the fact that the XFEL structures were obtained at not high resolutions (mostly at 2.8 Å). I consider that their systematic calculations may suggest a useful theoretical interpretation of the XFEL study.

Comments on latest version:

The authors have satisfied my concerns. I consider that their present manuscript is more attractive and informative for readers.

---

## [Author Response]

The following is the authors’ response to the previous reviews.

**Reviewer #1 Public Review**
“First, I agree with the authors of this manuscript that conformational changes in the XFEL structures with 2.8 A resolution are not reliable enough for demonstrating the subtle changes in the electron transfer events in this bacterial photosynthesis system. Actually, the data statistics in the paper by Dods et al. showed that the high-resolution range of some of the XFEL datasets may include pretty high noise (low CC1/2 and high Rsplit) so the comparison of the subtle conformational changes of the structures is problematic.The manuscript by Gai Nishikawa investigated time-dependent changes in the energetics of the electron transfer pathway based on the structures by Dods et al. by calculating redox potential of the active and inactive branches in the structures and found no clear link between the time-dependent structural changes and the electron transfer events in the XFEL structures published by Dods, R.et al. (2021). This study provided validation for the interpretation of the structures of those electrontransferring proteins.The paper was well prepared.”

Thank you very much for your positive and insightful comment. We greatly appreciate your suggestion regarding the high noise levels of the XFEL structures. Including this information in the Introduction section will draw readers’ attention to the concerns about the reliability of these XFEL structures. We have incorporated it into the Introduction section.

**Reviewer #2 Public Review**
“The manuscript by Nishikawa et al. addresses time-dependent changes in the electron transfer energetics in the photosynthetic reaction center from Blastochloris viridis, whose time-dependent structural changes upon light illumination were recently demonstrated by time-resolved serial femtosecond crystallography (SFX) using X-ray free-electron laser (XFEL) (Dods et al., Nature, 2021). Based on the redox potential Em values of bacteriopheophytin in the electron transfer active branch (BL) by solving the linear Poisson-Boltzmann equation, the authors found that Em(HL) values in the charge-separated 5-ps structure obtained by XFEL are not clearly changed, suggesting that the P+HL- state is not stabilized owing to protein reorganization. Furthermore, chlorin ring deformation upon HL- formation, which was expected from their QM/MM calculation, is not recognized in the 5ps XFEL structure. Then the authors concluded that the structural changes in the XFEL structures are not related to the actual time course of charge separation. They argued that their calculated changes in Em and chlorin ring deformations using the XEFL structures may reflect the experimental errors rather than the real structural changes; they mentioned this problem is due to the fact that the XFEL structures were obtained at not high resolutions (mostly at 2.8 Å). I consider that their systematic calculations may suggest a useful theoretical interpretation of the XFEL study. However, the present manuscript insists as a whole negatively that the experimental errors may hamper to provide the actual structural changes relevant to the electron transfer events.”

Thank you for your feedback on our manuscript. We appreciate your positive assessment of our systematic calculations and theoretical interpretation of the XFEL study. We have carefully considered your comments and made the necessary revisions to address your concerns.

**Reviewer #2 Recommendations for the authors**
“The authors have satisfied my concerns mostly, in particular by providing the Em(QA) changes, which seem to be more attractive in the present form. However, the Em(QA) value(s), at least in the dark structure, should be provided, and the procedure of the calculation for the Em(QA) value(s) should be described in METHODS "Calculation of Em".

The calculated Em(QA) values for dataset a and dataset b in the dark structure are –223 mV and – 209 mV, respectively, using the reference Em value of –256 mV versus NHE for menaquinone-2 in water [Photosynth. Res. 134 (2017) 193]. These calculated values are comparable to experimentally measured values of –150 mV for PbRC from Blastochloris viridis (naphtoquinone) [Biochim. Biophys.Acta 440 (1976) 622] and –180 mV for PbRC from Rhodobacter sphaeroides (ubiquinone) [Arch. Biochem. Biophys 172 (1976) 329].

We have now provided this information in the Method (“Calculation of Em”) and Results and Discussion (“Relevance of structural changes observed in XFEL structures”) sections.